# Emerging Epigenetic Targets and Their Molecular Impact on Vascular Remodeling in Pulmonary Hypertension

**DOI:** 10.3390/cells13030244

**Published:** 2024-01-28

**Authors:** A. Dushani C. U. Ranasinghe, T. M. Parinda B. Tennakoon, Margaret A. Schwarz

**Affiliations:** 1Boston Children’s Hospital, Harvard Medical School, Boston, MA 02115, USA; dushani.ranasinghe@childrens.harvard.edu (A.D.C.U.R.); parinda.tennakoon@childrens.harvard.edu (T.M.P.B.T.); 2Department of Pediatrics, Indiana University School of Medicine, 1234 Notre Dame Ave, South Bend, IN 46617, USA

**Keywords:** vascular remodeling, pulmonary hypertension, epigenetic, lung cancer

## Abstract

Pulmonary Hypertension (PH) is a terminal disease characterized by severe pulmonary vascular remodeling. Unfortunately, targeted therapy to prevent disease progression is limited. Here, the vascular cell populations that contribute to the molecular and morphological changes of PH in conjunction with current animal models for studying vascular remodeling in PH will be examined. The status quo of epigenetic targeting for treating vascular remodeling in different PH subtypes will be dissected, while parallel epigenetic threads between pulmonary hypertension and pathogenic cancer provide insight into future therapeutic PH opportunities.

## 1. Introduction

Pulmonary hypertension (PH) is a progressive and incurable cardiopulmonary vascular disease that is heterogeneous and multifactorial in nature. In PH, a delayed diagnosis is associated with diagnostic challenges due to the late onset of symptoms, epidemiological, sex-related, and pathophysiological differences [1]. Recently, the diagnostic mean pulmonary artery pressure (mPAP) threshold was lowered from >25 mm Hg to >20 mm Hg as studies suggest that lowering pulmonary vascular resistance standards from ≥3.0 Wood units (WU) to >2.0 WU is warranted to promote early identification of PH patients, with the goal of reducing long-term disease burden [2]. Discovering therapeutic approaches to manage PH vascular remodeling is critical to meeting this target and enhancing PH patients’ quality of life.

As the evolution of PH is multifactorial, encompassing exposure to pulmonary vascular insults such as genetic mutations, hypoxia, drugs, pathogens, and environmental pollutants, etc., the challenge is in identifying specific treatable targets. Therefore, it is vital that we understand not only key triggering factors that initiate vascular remodeling, but also the growing list of cellular phenotypes that contribute to this aberrant process. Here, we discuss the molecular and morphological reprogramming of different vascular cell populations, the pathogenic pathways that resemble cancer, and their epigenetic impact on pulmonary vascular remodeling in PH with a particular emphasis on group 1 PH, also named as pulmonary arterial hypertension (PAH). Vascular cell populations in this review include endothelial cells (ECs), smooth muscle cells (SMCs), pericytes, fibroblasts, and myofibroblasts in different experimental PH animal models.

## 2. Epigenetics

Epigenetic modifications encompass any process that results in a change in the gene expression pattern without external factors altering the genetic code. Highly organized, eukaryotic DNA are tightly packaged in nucleosomes with histones in the nucleus. Epigenetic chromatin remodeling modifications occur mainly via three mechanisms including (1) DNA/RNA methylation, (2) histone post-translational modifications, and (3) an RNA interference through non-coding RNAs (ncRNAs), including small (microRNAs (miRNAs), small interfering RNAs (siRNAs), Piwi-interacting RNAs (piRNAs), and long noncoding RNAs (lncRNAs), resulting in the activation or silencing of genes. DNA methylation, defined by the addition of a methyl group (CH3) to cytosine residues to form 5-methylcytosine with DNA methyltransferases (DNMTs), usually turns off gene expression by facilitating chromatin condensation. Importantly, the histone octamer that DNA is wound around consists of dimers of core histones H2A, H2B, H3, and H4. Within the histone core, the N- and C-terminal amino acid tails of the histone core can undergo post-translational modifications including methylation, acetylation, phosphorylation, sumoylation, ubiquitylation, and ribosylation, resulting in the modulation of DNA accessibility by providing binding platforms for transcription factors (TFs) and DNA/chromatin-modifying or remodeling enzymes. An altered epigenetic landscape may drive PH disease pathogenesis [3].

## 3. Epigenetics in Transcriptional Regulation

The regulation of gene expression lies at the core of cellular and molecular phenotype changes. The epigenetic processes, such as DNA methylation, histone modification, and various RNA-mediated mechanisms, can influence gene expression mainly at the level of transcription. Regardless of the underlying cause, chromatin remodeling using these epigenetic mechanisms, activates or inhibits specific genes through alterations in regulatory regions. This process, whether it involves the activation of previously silent genes or the deactivation of crucial maintenance genes, can initiate and contribute to disease development. For example, DNMTs are involved in de novo DNA methylation and maintain global genomic methylation that usually turns off gene expression by promoting chromatin condensation. Similar to DNA methylation, histone methylation is the process by which multiple methyl groups are added to lysine (K) residues or arginine (R) residues of histone proteins with histone methyltransferases (HMTs). Here, methylation of histones can either activate or inhibit gene expression, either by loosening chromatin tails or by promoting tightening of chromatin structures thus regulating the access of transcription factors and other proteins to DNA. Conversely, histone lysine acetylation is typically linked to the activation of gene transcription. In this process, acetylation of histones neutralizes the positive charge of the amino acid, loosening the interaction with DNA and promoting an ‘open’ chromatin structure, thereby facilitating gene transcription. Additionally, acetylated histones attract ‘readers’, including bromodomain (BRD) and extra-terminal (BET) proteins, which recruit transcriptional machinery and other chromatin-modifying elements through hydrogen bonding. Interestingly, the expression patterns of ncRNAs can be cell- or tissue-type specific and demonstrate cell-type-specific mechanisms and phenotypes [1]. Understanding the forces that modulate epigenetic mediated gene transcription in PH vascular remodeling can facilitate the development of therapeutic targets.

## 4. PH Disease Models

One of the major challenges in the PH research field is the availability of a simple and severe PH animal model. Multiple animal models have been designed and utilized to investigate the pathobiology, molecular, and mechanistic insights of PH. Commonly used experimental PH models in epigenetics studies include chronic hypoxia and Sugen hypoxia and Monocrotaline models [2,3,4,5], and future explorations should be warranted in gene editing and pneumonectomy models as well [4].

Chronic hypoxic modeling causes PH-associated pulmonary vascular remodeling with decent predictability and repeatability. However, most of these vascular changes can be restored by returning to a normoxic environment. A combination of the chronic hypoxia model with the vascular endothelial growth factor receptor (VEGFR)-2 antagonist Sugen 5416 (SU5416; semaxanib) (SuHx) results in more severe remodeling and the development of plexiform lesions that are more resistant to reversal under a normoxic environment. These hypoxic models, however, can have gender, strain, and age differences [5].

Monocrotaline (MCT), an 11-membered macrocyclic pyrrolizidine alkaloid that is metabolized into the toxic metabolite MCT pyrrole (MCTP) by liver cytochrome P450 3A4 (CYP3A4), results in vascular EC damage and inflammation-inducing PH, associated RV hypertrophy, and an increase in the medial thickness of pulmonary arteries in rats. Although the MCT model has low cost, simplicity, and reproducibility, its broad toxicity causes this model to be less in line with human PH [6].

More severe phenotypes in PH can be achieved by imposing secondary vascular insults such as chronic hypoxia or an MCT challenge to gene editing models such as BMPR2 mutants and IL6 knockouts, while a left/right pneumonectomy model with MCT or hypoxia has been utilized to model flow-induced PH. Moreover, a recent study suggests that an extended pneumonectomy, achieved by simultaneous removal of the left lung and right caval lobe in mice, is a promising animal model to study the cellular response and molecular mechanisms contributing to compensatory lung growth and flow-induced PH [7]. In addition to these models, bleomycin and Schistosome treatment can induce PH in rodents [4].

## 5. Cellular Epigenetic Changes

There are multiple vascular cell types known to be involved in the pathogenesis of PH including endothelial cells, smooth muscle cells, fibroblasts, and pericytes.

### 5.1. Endothelial Cells

The most inner wall of the blood vessel is lined with a monolayer of endothelial cells (ECs). This EC monolayer is responsible for the flux and exchange of various substances during circulation into the parenchymal tissue, preservation of the vascular barrier, maintenance of vascular tone, thrombosis prevention, inflammation, and modulation of neighboring mural cells including smooth muscle cells and pericytes. Early in the onset of PH, following vascular insults, damaged ECs are triggered to undergo apoptosis. With PH progression, apoptosis-resistant ECs emerge [1], and eventually in later stages, ECs become senescent resulting in a non-reversible dysfunctional endothelium that contributes to neointima formation and endothelial to mesenchymal transition (EndoMT). Together this contributes to the progression of PH as there is less protection against thrombosis, leakage, and inflammation.

Multiple epigenetic mechanisms have been implicated in PH endothelial dysfunction [6,7,8,9,10]. Extensive remodeling of the active enhancer landscape with an acetylated histone H3K27 mark was reported in pulmonary arterial endothelial cells (PAECs) derived from PH patients [8]. Moreover, H3K27 acetylation required for EC regeneration was absent in PH. In other studies, histone H3K9 acetylation-mediated BOLA3 deficiency, increased endothelial proliferation, survival, and vasoconstriction while leading to decreased angiogenic potential in multiple PH models including hypoxic mice. In contrast, the mimicking of bromodomain and extra-terminal (BET) proteins that bind to acetylated histones was found to decrease inflammation and cell cycle progression of pulmonary ECs in PH [9]. Epigenetic enzymes, such as SIRT3, can modulate EC metabolism via the acetylation of non-histone proteins [10]. DNA methylation in EC was previously shown to modulate endothelial NO synthase (eNOS) activity via the methylation of a proximal promoter of eNOS resulting in interrupted vasodilation and endothelial homeostasis. Moreover, a previous DNA methylation analysis in pulmonary ECs of PH patients identified a set of genes mainly involved in the lipid transport pathway that could be relevant to PH pathophysiology [11]. Other studies found that N6-methyladenosine (m6A)-modified transcripts of lncRNAs have a role in Pyroptosis (a form of programmed cell death) in ECs in hypoxic mediated PH via DNA methylation. Thus, demonstrating a collaboration of multiple epigenetic modifications [12]. Lastly, a recent study showed that long non-coding RNA growth arrest-specific transcript 5 (GAS5) promoted spermidine (SP)-induced autophagy in pulmonary ECs in PH and in a hypoxia rat model by targeting miRNA-31-5p [13]. The therapeutic potential of microRNAs has been assessed in multiple studies including miR-150 supplementation that demonstrated an attenuation in pulmonary endothelial damage induced by vascular stresses [11,12,13,14]. Further studies are needed to better understand the epigenetic influence and influencers of endothelial dysregulation.

### 5.2. Smooth Muscle Cells

Vascular smooth muscle cells (SMCs), the medial layer of arteries, are significant contributors to maintaining the integrity of structure and function in the pulmonary vessels. Extensive pulmonary vascular remodeling in a PH lung is mainly seen in the small to mid-size arteries, <500 μm in diameter, that possess a significant thickening of media. Importantly, SMC can reversibly undergo phenotypic switching, where it can go from a quiescent to a contractile or synthetic phenotype while acquiring or losing proliferative and migratory potential. Moreover, during phenotypic switching, the capacity to synthesize the extracellular matrix (ECM) is altered, while single-cell transcriptomic studies show the significance of heterogeneity of SMCs in remodeled PAs [15]. Thus, a conceptual relationship between PH vascular remodeling and pathologic cancer theory, supported by pulmonary SMC hyperproliferation, resistance to apoptosis, and increased migration [1], continues to gain traction.

Accumulating studies show the role of epigenetics in SMC phenotypic switching or transition from contractile to synthetic, with increased proliferative and migratory capacities. An epigenetic modifier, Switch-independent 3a (SIN3a) overexpression decreased the HDAC activity and methylation level of the BMPR2 promoter attenuating SMCs’ hyperproliferation and migration in PH in both patients’ tissues and MCT and SuHx PH models [16]. Furthermore, recent studies define an important role for histone acetylation in PH-SMCs vascular remodeling. For example, increased levels of aldehyde dehydrogenase, ALDH1A3 (aldehyde dehydrogenase family 1 member 3) promote acetyl coenzyme A to acetylate histone H3K27, while inducing the highly proliferative and glycolytic pulmonary artery SMC (PASMC) in PH [17]. Histone H3K9 acetylation was significantly increased in the lungs of PH patients and promoted PASMC proliferation via Sphingosine Kinase 2 [18]. CircRNA, Hsa_circ_0001402 acts as an miR-183-5p sponge to inhibit SMC proliferation and migration while activating VSMC autophagy to alleviate neointimal hyperplasia [19]. Another SMC epigenetic target is SETD2; the main enzyme that catalyzes the trimethylation of H3K36 (H3K36me3) as SMCs-specific SETD2 deficiency ameliorated the pathological pulmonary vascular remodeling in a hypoxia-induced mouse model of PH [20]. Moreover, emerging studies suggest that pulmonary ECs can undergo EndoMT transition, resulting in mesenchymal/smooth muscle-like cells in PH. Although these studies support a broader role for epigenetics in SMC phenotypic switching, factors regulating this process and their downstream molecular targets require further exploration.

### 5.3. Adventitial Fibroblasts

Fibroblasts are the most common cell type in the adventitial layer. The activated fibroblast, also known as a myofibroblast, is hyperproliferative, synthesizes increased ECM, and secretes inflammatory cytokines.

Recent studies determined that fibroblasts from PH patients treated with combined therapy of sildenafil (vasodilator) and HDAC inhibitor exhibited synergistic inhibitory effects on PH-fibroblast proliferation and induced metabolic reprogramming [21]. A recent study in multiple PH models suggests that pharmacological inhibition of the P300/CREB-binding transcriptional co-activators and histone acetyl transferase (HAT) complex rescues distal pulmonary vascular remodeling and hemodynamics in PH models as well as the vascular remodeling in precision-cut tissue slices from human PH lungs ex vivo [22]. Other studies demonstrated, via integrative analyses of RNA-seq and ChIP-seq data of PH-fibroblasts, that the altered epigenetic landscape signatures of fibroblasts in PH were similar to those found in lung morphogenesis [22] highlighting a reactivation of developmental pathways in PH progression, while the role of miR-124 in PH fibroblasts has been demonstrated to modulate proliferative, migratory, and inflammatory phenotypes of fibroblasts in PH [23]. These findings support the importance of expanding our understanding of the epigenetic factors that reawaken latent lung development tendencies and their contribution to the development of PH.

### 5.4. Pericytes

Pericytes provide structural support for the endothelial tube and participate in vascular tone maintenance [24,25]. Accumulation of pericytes in the distal pulmonary arteries is seen in PH patients [26,27], with single-cell transcriptomics findings supportive of the recruitment of pericytes to the inflamed pulmonary arteries in PH and the potential of pericytes to differentiate into smooth muscle-like cells [26,27]. Emerging studies implicate a role of pericytes in PH, where tyrosine kinase receptor-inducing lncRNA and TUG1 lncRNA induce a pro-proliferative phenotype in PH [26,28,29]. Unfortunately, broadening our understanding of epigenetic factors on pericyte contribution to PH vascular remodeling is challenged by a lack of consistent molecular markers and the subsequent difficulties associated with distinguishing pericytes from other cell types. However, recent advancements in multiomics, single-cell and spatial transcriptomics, CUT&RUN, and CUT&TAG technologies will likely improve pericyte research in PH.

### 5.5. Cell–Cell Interactions

Single-cell transcriptomic data demonstrate that the cell–cell communication in PH is significantly altered and transforms towards SMC–fibroblasts contacts [15]. Recent studies highlight the importance of SMC–EC contact for EC regeneration. Importantly, this process is mediated by epigenetic histone acetylation [30]. Other studies determined that endothelial-secreted factors promote the hyperproliferation of SMCs using epigenetic mechanisms such as histone acetylation and microRNA [18,31]. Future exploration is needed to further determine the epigenetic contribution that cell–cell interaction has to PH vascular remodeling and to assess the reversibility of epigenetic targeting in PH (Figure 1).

## 6. Vascular Remodeling: Unveiling Common Epigenetic Threads in Pulmonary Hypertension and Associated Proliferative Diseases Such as Lung Cancer

Parallels between numerous respiratory disorders, particularly lung cancer (LC), chronic obstructive pulmonary disease (COPD), and idiopathic pulmonary fibrosis (IPF), have long piqued the interest of researchers. Notably, in 1998 PH was linked to malignant disorders due to its pathogenic pathways resembling some elements of cancer [32], with recent studies revealing significant genetic and epigenetic overlap between different types of LC and PH [33,34,35].

The similarity between cancer behaviors and some disease characteristics of PH was a startling discovery with PH PASMCs and ECs exhibiting cellular traits typical of cancer including hyper-proliferation and resistance to apoptosis [33,34,35]. In contrast to tumor cell hyperproliferation, aberrant proliferation in PH causes vascular remodeling and contributes considerably to the unique symptoms of PH of pulmonary artery constriction and increased pulmonary vascular resistance [36]. Several studies have found a strong link between PH and LC [37,38,39,40], as a considerable number of LC patients also develop PH [37,38,41]. In LC, tumor epithelial cells affect the tumor microenvironment, causing vascular remodeling and contributing to the establishment of PH. This association has been corroborated across various mouse models, underscoring the intricate interplay between LC epithelial cells, SMC, and ECs. Emerging relationships between the tumor microenvironment and vascular remodeling suggest that LC and PH may share common epigenetic mediators and warrant future exploration.

The development of PH in LC patients creates complications, possibly affecting surgery outcomes and, in extreme situations, hindering resection feasibility [41]. Despite recent advances in understanding the frequency of both LC and PH, the complicated processes underlying their relationship remain unknown. However, the presence of PH in LC patients provides an opportunity to use surrogate PH indicators to predict LC prognosis. Consequently, an in-depth investigation of the development of PH in LC patients is required, with the goal of using it as diagnostic and prognostic indicators. Aligned with the central theme of this review focusing on PH and epigenetics, this segment will focus on the epigenetic factors influencing the concurrent occurrence of PH in LC patients. Additionally, this review considers the role that employing these epigenetic markers has for novel diagnostic and prognostic applications for PH management in LC patients.

SMCs and ECs play critical roles in PH [33,35,36]. To investigate epigenetic abnormalities in LC patients that may contribute to the development of PH, a targeted analysis of epigenetic variables impacting these particular cell types within the lung environment is required. However, the relationship between LC epithelial cells and their cellular counterparts is complex. As epigenetic modifications in LC epithelial cells may cause changes in the chemicals produced by tumor cells, potentially training SMCs and ECs to be hyperproliferative and apoptosis-resistant, leading to PH. Epigenetic changes inside SMCs and ECs, mediated by interactions with tumor epithelial cells, may, on the other hand, induce such traits in these cell types. Understanding the epigenetic remodeling that leads to PH in LC patients requires a holistic approach to both illnesses.

Tumor epithelial cells release cytokines, chemokines, and other growth factors to alter the tumor microenvironment and promote tumor proliferation and migration [42,43]. Driven in part by the tumor microenvironment that contains a diverse range of stromal and immune cell types [37,44,45,46]. By hijacking the immune system, tumors create an immunosuppressive environment to avoid immune-cell-mediated tumor cell death, and more crucially, to modify the vasculature to maintain the input of oxygen and nutrients to continue growing [47,48,49]. To induce angiogenesis, tumor epithelial cells release angiogenic factors such as Vascular endothelial growth factor (VEGF) and Platelet-derived growth factor (PDGF). Given their function in vascular remodeling, plasma levels of these factors are understandably higher in patients with severe PH [50,51]. As a result, focusing on the control of the angiogenic factors of lung tumor cells, as well as how they are altered by epigenetic alterations, may help us understand the common incidence of PH in LC patients. This section will attempt to unravel how lung tumor epithelial cells alter the lung vasculature to increase PH via epigenetic modifications while addressing whether these epigenetic markers can be utilized to predict the incidence of PH in LC patients and be targeted to treat PH in LC patients (Figure 2).

Posttranslational histone modifications, DNA changes, and microRNAs (miRNAs) are the most prevalent epigenetic alterations used by lung tumor cells to induce angiogenesis and modify the vasculature [52,53]. Histone acetylation and methylation are the two most frequent epigenetic changes that influence angiogenesis in LC [52] with Histone deacetylases (HDACs), particularly HDAC1, playing a key role in vascular integrity [54]. Although research on HDACs and angiogenesis in LC is scarce, their impact on angiogenesis in other cancers [55,56] suggests a potential role for HDACs in LC-mediated vascular modifications. Another epigenetic alteration that controls LC angiogenesis is DNA methylation. Studies indicate that the methylation state of a promoter regulates genes critical to vascular remodeling such as VEGF expression [57]. MicroRNAs are another epigenetic regulator that affects angiogenesis in LC. MiRNAs, short single-stranded non-coding RNAs with total lengths ranging from 19 to 25 nucleotides, exert control over gene expression by triggering translational repression via fast mRNA degradation with deadenylation. MicroRNAs that target angiogenic factors such as VEGF-A (miR-128, miR-200b) are downregulated in [58,59,60], but MicroRNAs that target anti-angiogenic factors (miR-221/222 cluster, miR-210) are elevated. These epigenetic alterations imply that PH maybe linked to lung cancer.

### Implications of LC Associated PH Management

The relationship between angiogenesis in LC and probable PH raises an essential question: can we use these common epigenetic modifications in LC patients as diagnostic tools for PH? The key to unraveling this conundrum is determining how these modifications that occur in tumor epithelial cells, SMCs, and ECs, may be targeted for PH treatment in LC. Despite scant research on specific epigenetic alterations in LC-related PH, investigating shared changes in both disorders provides insights into possible indicators and therapeutic methods for this condition.

Several studies suggest that blood VEGF levels can be used to determine the severity of LC. Interestingly, VEGF plasma levels are also increased in patients with severe PH [61,62], with VEGF and VEGF receptor 2 (VEGFR2) being strongly expressed in complicated vascular lesions in the lungs of PH patients [63]. Taken together, this might be considered a realistic alternative for predicting LC-linked PH, while targeting epigenetic markers, in addition to anti-angiogenic medications, could be a viable strategy for preventing/treating PH in these patients. Histone acetylation, as previously established, is one of the major epigenetic alterations that governs angiogenesis in LC. Surprisingly, it is a modification that occurs often in PH as well [18,63,64]. Unfortunately, medications that inhibit HDAC (HDACi) such as Vorinostat (Zolinza), Romidepsin (Istodax), and Belinostat (Beleodaq), already approved by the FDA as safe for use in certain lymphoma cases, lack conclusive data regarding their efficacy in treating solid tumors or whether tumors with specific genetic changes may respond better to these drugs [64,65,66]. Nonetheless, considering the importance of HDACs and HATs in reprogramming vasculature in LC, they may have substantial promise as a therapy not just for LC, but also for LC-linked PH. As mentioned previously, DNA methylation is another significant epigenetic alteration in LC that affects vascular remodeling. Like histone acetylation, DNA methylation is also seen extensively in PH [67,68]. DNA methylation can be inhibited using DNA methyltransferase inhibitors (DNMTis) [69]. Unfortunately, there are no FDA-approved DNMTis for the treatment of LC. Azacitidine and decitabine, both DNA methyltransferase inhibitors, have undergone clinical studies in several malignancies, including LC, to determine their efficacy and safety [70]. Their approvals include myelodysplastic syndromes (MDS) and acute myeloid leukemia (AML) [71], but not have been approved for the treatment of any solid tumors. Based on the function of DNA methylation and vascular remodeling, these inhibitors may influence the epigenetic pathways involved in PH development, thereby opening a novel treatment route in LC-associated PH. To alleviate PH, they might act on both tumors and cells responsible for vascular remodeling. Finally, like HDACis and DNMTis, no miRNA mimics or inhibitors are available for the therapy of solid malignancies. Furthermore, miRNA-related medicines are still in the early stages of research for a variety of disorders. Of note, four novel RNAi-based therapies have been authorized by the FDA: patisiran, givosiran, lumasiran, inclisiran, and Vutrisiran [72,73,74,75]. Each of these siRNA medicines targets distinct mRNA transcripts to treat conditions other than cancer. The epigenetic mechanisms contributing to the development of PH in LC patients are outlined in Figure 2. As a result, numerous advances in the use of miRNA-related medicines for the treatment of LC-related PH still need to be made.

Taken as a whole, the holistic review of the existing literature reveals that, while addressing epigenetic changes has the potential for treating LC-related PH, significant research in this area is absent. Notably, using epigenetic targeting in conjunction with standard anti-angiogenic medications to target angiogenesis in LC may be promising. Because LC and PH share epigenetic changes, focusing on these similarities has a double advantage. Targeting these changes might effectively address the complex interaction between lung tumor epithelial cells, ECs, and SMCs, thereby preventing the development of pulmonary hypertension in LC patients.

## 7. Conclusions

Epigenetic mediators impact PH vascular remodeling through molecular and morphological reprogramming of ECs’, SMCs’, pericytes’, fibroblasts’, and myofibroblasts’ vascular cell populations (Figure 3), with activated pathogenic pathways not only resembling cancer, but also affording valuable insight into future therapeutic opportunities. Taken together, these studies support the importance of the identification of PH epigenetic mediators and their targets. 

## Figures and Tables

**Figure 1 cells-13-00244-f001:**
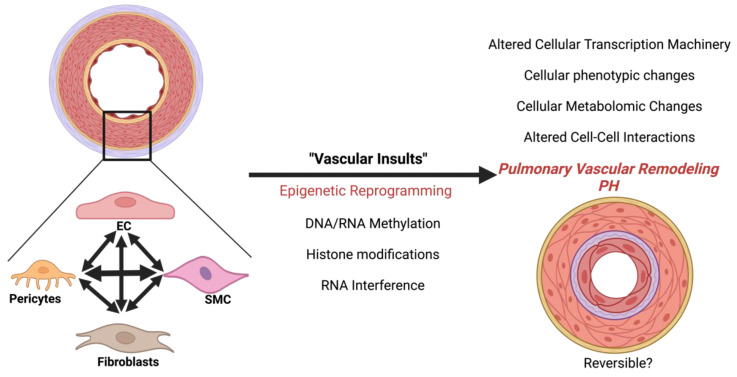
The progression of pulmonary hypertension via epigenetic reprogramming.

**Figure 2 cells-13-00244-f002:**
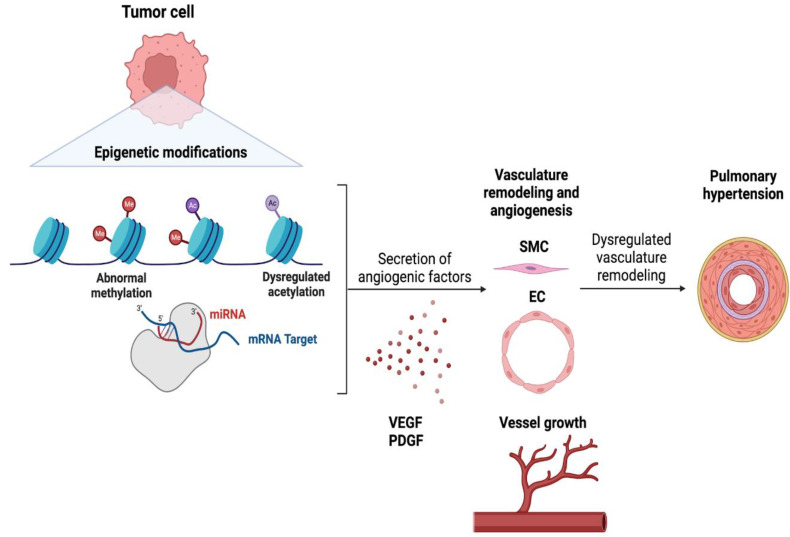
Role of epigenetic reprogramming in contemporary pulmonary hypertension in lung cancer.

**Figure 3 cells-13-00244-f003:**
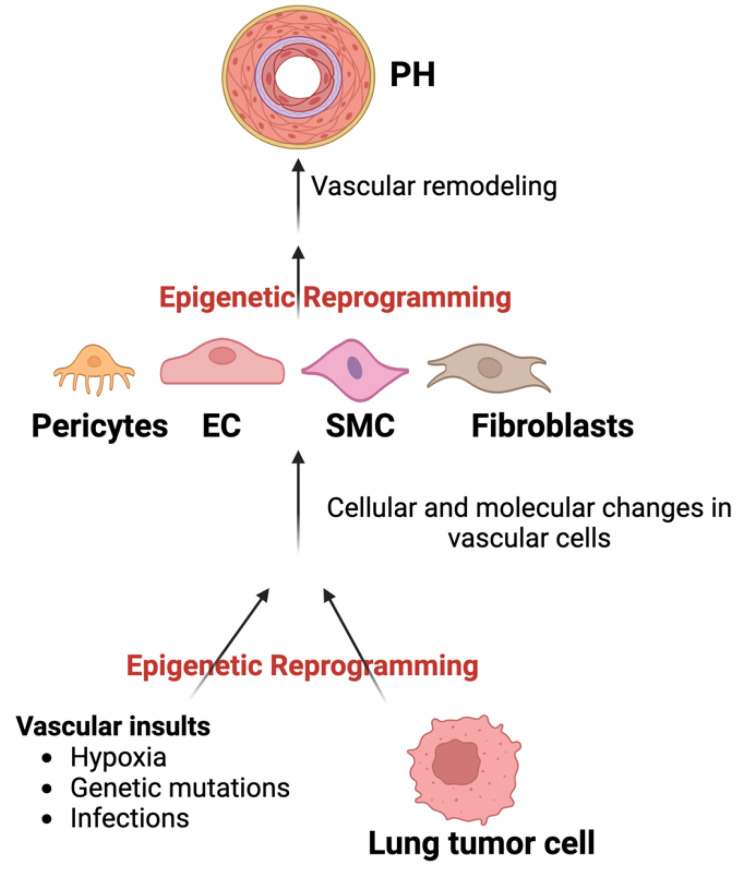
Proposed epigenetic interface between LC and PH vascular remodeling.

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
