# Peer review of "Emerging Epigenetic Targets and Their Molecular Impact on Vascular Remodeling in Pulmonary Hypertension"

_cells, 2024, doi:10.3390/cells13030244_

Round 1

Reviewer 1 Report

Comments and Suggestions for Authors

This comprehensive review is well-written, scientifically sound, and interesting. It summarizes the epigenetic mechanisms in pulmonary vascular cells and the molecular impact on vascular remodeling in pulmonary hypertension.  In addition, the authors added an interesting section on the link and how lung cancer alters the lung vasculature to increase PH via epigenetic modifications.

1. I would suggest adding a short chapter to discuss the effects and mechanism by which histone modifications and DNA methylation affect transcriptional regulation.

2. Recommend double-checking PAH and PH throughout the text.

3. I also suggest adding an abbreviations section or double-checking non-standard terminology.

Comments on the Quality of English Language

Good quality of Enlglish

Author Response

This comprehensive review is well-written, scientifically sound, and interesting. It summarizes the epigenetic mechanisms in pulmonary vascular cells and the molecular impact on vascular remodeling in pulmonary hypertension. In addition, the authors added an interesting section on the link and how lung cancer alters the lung vasculature to increase PH via epigenetic modifications.

Thank you for recognizing the importance of our manuscript.

  1. I would suggest adding a short chapter to discuss the effects and mechanism by which histone modifications and DNA methylation affect transcriptional regulation.

This is a very good point and now, we have included a short chapter with the mechanisms of DNA methylation and histone modifications in the regulation of transcription machinery.

  1. Recommend double-checking PAH and PH throughout the text.

Sorry for any confusions. We now, have corrected this.

  1. I also suggest adding an abbreviations section or doublechecking non-standard terminology

We now, have included a section with non-standard abbreviations.

Reviewer 2 Report

Comments and Suggestions for Authors

Dear Authors.

Your chosen topic for the manuscript review is quite interesting. The structural organization of ideas in the sections appears appropriate, following a logical order.  The bibliography predominantly comprises recent sources, providing substantial support for the development of concepts.

However, upon reviewing each section, it becomes evident that some citations necessary for substantiating the flow of ideas are missing. For instance, in the section discussing PH models, each paragraph introduces a specific model, but the depth of exploration is lacking. Your manuscript focuses on epigenetic therapeutic targets, yet the discussion of these models remains superficial.

While the delineation of different cell types is accurate, the content lacks adequate citation support, diminishing the impact of each paragraph. Many ideas within paragraphs are often supported by a single reference, indicated by phrases such as "multiple X...," leading to ambiguity and leaving the impression of a superficial review.

The sections comparing PH with cancer are intriguing; however, similar observations regarding citation depth persist. Furthermore, these sections lack a clear summation of the primary ideas or a conclusion. It would greatly benefit the manuscript to include a summary model or a figure encapsulating the central concepts concerning epigenetic therapeutic targets.

Comments on the Quality of English Language

There is a good use of English grammar and language.

Author Response

Your chosen topic for the manuscript review is quite interesting. The structural organization of ideas in the sections appears appropriate, following a logical order. The bibliography predominantly comprises recent sources, providing substantial support for the development of concepts.

Thank you for appreciating our work.

However, upon reviewing each section, it becomes evident that some citations necessary for substantiating the flow of ideas are missing. For instance, in the section discussing PH models, each paragraph introduces a specific model, but the depth of exploration is lacking. Your manuscript focuses on epigenetic therapeutic targets, yet the discussion of these models remains superficial.

We apologize for the confusion. The PH models we discussed were the models that were mostly used in epigenetics research in PH. We have now clarified this idea in the manuscript.

While the delineation of different cell types is accurate, the content lacks adequate citation support, diminishing the impact of each paragraph. Many ideas within paragraphs are often supported by a single reference, indicated by phrases such as "multiple X...," leading to ambiguity and leaving the impression of a superficial review.

We have included necessary citations for these instances. Thank you for noticing it.

 The sections comparing PH with cancer are intriguing; however, similar observations regarding citation depth persist. Furthermore, these sections lack a clear summation of the primary ideas or a conclusion. It would greatly benefit the manuscript to include a summary model or a figure encapsulating the central concepts concerning epigenetic therapeutic targets.

We appreciate the reviewer's comments on references in the section discussing the relationship between lung cancer and pulmonary hypertension. We are glad that the reviewer found this section interesting. Regarding the concern about the limited citations of peer-reviewed studies in this section, we acknowledge that the current literature on epigenetic modifications in the context of lung cancer and pulmonary hypertension is notably limited and small in number. We want to underscore the uniqueness of our concept, which involves the potential utilization of epigenetic modifications to predict and possibly target the reduction of pulmonary hypertension occurrence in lung cancer patients. It's worth noting that the scarcity of available references may be attributed to the novelty of this idea, making it an emerging area of exploration in the field.

Our manuscript introduces a novel idea, and we aimed to exhaustively search for any studies that either support or challenge this claim. Despite the limited existing literature, our intention is to highlight the groundbreaking nature of this concept. The absence of conclusive evidence in the current studies underscores the need for further exploration in this area. We believe that our article, by bringing this novel idea to light, lays the foundation for future studies that may delve deeper into the relationship between epigenetic modifications and the co-occurrence of lung cancer and pulmonary hypertension. This concept, while speculative at this stage, has the potential to shape the direction of future research in the field.

We have now added a summary figure.

Round 2

Reviewer 2 Report

Comments and Suggestions for Authors

Dear authors.

Thank you very much for insisting on and improving the submitted manuscript. I appreciate that you have included my comments, and rethought some of my suggestions.  I find that this version has the form and content to be approved.

Congratulations